# Effect of Layered Aminovanadic Oxalate Phosphate on Flame Retardancy of Epoxy Resin

**DOI:** 10.3390/molecules28083322

**Published:** 2023-04-09

**Authors:** Po Hu, Weixi Li, Shuai Huang, Zongmian Zhang, Hong Liu, Wang Zhan, Mingyi Chen, Qinghong Kong

**Affiliations:** School of Emergency Management, Jiangsu University, Zhenjiang 212013, China

**Keywords:** hydrothermal method, layered structure, phosphate, flame retardancy, vanadium, epoxy resin

## Abstract

To alleviate the fire hazard of epoxy resin (EP), layered ammonium vanadium oxalate-phosphate (AVOPh) with the structural formula of (NH_4_)_2_[VO(HPO_4_)]_2_(C_2_O_4_)·5H_2_O is synthesized using the hydrothermal method and mixed into an EP matrix to prepare EP/AVOPh composites. The thermogravimetric analysis (TGA) results show that AVOPh exhibits a similar thermal decomposition temperature to EP, which is suitable for flame retardancy for EP. The incorporation of AVOPh nanosheets greatly improves the thermal stability and residual yield of EP/AVOPh composites at high temperatures. The residue of pure EP is 15.3% at 700 °C. In comparison, the residue of EP/AVOPh composites is increased to 23.0% with 8 wt% AVOPh loading. Simultaneously, EP/6 wt% AVOPh composites reach UL-94 V1 rating (t_1_ + t_2_ =16 s) and LOI value of 32.8%. The improved flame retardancy of EP/ AVOPh composites is also proven by the cone calorimeter test (CCT). The results of CCT of EP/8 wt% AVOPh composites show that the peak heat release rate (PHHR), total smoke production (TSP), peak of CO production (PCOP), and peak of CO_2_ production (PCO_2_P) decrease by 32.7%, 20.4%, 37.1%, and 33.3% compared with those of EP, respectively. This can be attributed to the lamellar barrier, gas phase quenching effect of phosphorus-containing volatiles, the catalytic charring effect of transition metal vanadium, and the synergistic decomposition of oxalic acid structure and charring effect of phosphorus phase, which can insulate heat and inhibit smoke release. Based on the experimental data, AVOPh is expected to serve as a new high-efficiency flame retardant for EP.

## 1. Introduction

Epoxy resin (EP) is a class of polymer composed of large chains of hydrocarbons [1]. It has received much attention because of its excellent performance and convenient production. With the progress of technology, EP has been widely used in all aspects of production and life, such as coatings, architectural decoration, structural elements, and adhesives [2,3,4]. In the past decade, the global production of epoxy resin increased from 2.96 million tons in 2017 to 3.73 million tons in 2021, with a compound annual growth rate of 5.95%. As of 2021, the global total production capacity of epoxy resin was 5.37 million tons. However, EP also has disadvantages that limit its engineering applications. Due to the hydrocarbon structure of EP, it is flammable, resulting in instability at high temperatures, and easily cracks and burns [5,6]. At the same time, it releases a lot of heat and toxic gases, which could threaten life and property during a fire. To improve the safety of EP used in high-temperature environments and reduce its fire hazard, it is necessary to improve the flame retardancy of EP [7].

According to previous studies, two-dimensional (2D) layered phosphates have proven to be effective phosphate-containing flame retardants for EP [8,9,10]. Phosphate-based flame retardants are considered the most promising candidate to replace halogen-based flame retardants. Phosphate compounds have excellent flame retardancy in both gaseous and condensed phases. In the condensed phases, phosphorus is initially decomposed into phosphoric acid or dehydrated to form metaphosphate, which is polymerized to form polymetaphosphate. It has a catalytic effect on the dehydration of oxygen-containing polymers into char. In the gaseous phases, it acts as a free radical trapping agent that forms PO during combustion, which can combine with H· in the flame region to inhibit flame [11,12]. Materials with a 2D layered structure can provide a physical barrier effect, which can delay decomposition when they are added to the EP matrix. Phosphates containing transition metals, such as aluminum diethyl phosphate (ADP), copper phenylphosphate, and phosphor metal complex (CePn), are oxidized to form metal oxides at high temperatures [13,14,15,16]. Salen-diphenyl chloro-phosphate (Salen-DPCPs) metal compounds with phosphate structure and nickel components can reduce the flame hazards of EP. The results showed that Salen-DPCPs exhibited a good flame retardant effect for EP. The limiting oxygen index (LOI) value was up to 31.5%, and EP composites achieved a UL-94 V0 rating when 5 wt% Salen-DPCP-1 was incorporated [17]. It led to excellent catalytic properties, which accelerated cross-linking of EP to improve the charring properties [18,19]. Therefore, the combination of phosphorus and 2D metal compounds can reach a good synergistic effect, enhancing the compactness of the char layer.

The synergistic effect of nitrogen and phosphorus has a phosphorus-nitrogen structure-activity relationship, diluting the concentration of combustibles and oxygen around the combustion substrate and promoting flame extinction [20,21]. The results indicated that adding 7 wt% nitrogen/phosphorus modified lignin to epoxy resin achieved UL-94 V0 grade and limiting oxygen index of 31.4%, respectively. In addition, the residual char became denser [22]. NiNH_4_PO_4_·H_2_O nanoflakes have good flame retardancy in EP. When the addition amount was 5%, the peak heat release rate and peak smoke production of the composites were 69.1% and 36.5% lower than those of pure EP, respectively. NiNH_4_PO_4_·H_2_O promoted the formation of a stable carbon layer and released noncombustible gases. It prevented heat and oxygen transfer and diluted the concentration of combustible gases [21].

Inspired by the above work, 2D metal phosphonate with a layered structure and optimal thermal stability can effectively ameliorate the fire retardation and smoke inhibition performances of polymer composites. Additionally, the excellent synergistic flame retardant capability of gaseous phase flame retardancy, catalytic carbonation, and physical barrier can be achieved. In this work, ammonium vanadium oxalate-phosphate (AVOPh, (NH_4_)_2_[VO(HPO_4_)]_2_(C_2_O_4_)5H_2_O) was successfully designed and synthesized, which was incorporated into an EP matrix as flame retardant filler for preparing EP/AVOPh composites. To better understand the effect of various components of AVOPh on the flame retardancy of EP composites, ammonium vanadium phosphates (AVPh, (NH_4_)VOPO_4_·1.5H_2_O) and vanadium phosphates (VPh, VOPO_4_·2H_2_O) were selected as comparative samples. The results indicated that EP/AVOPh composites exhibited excellent thermal stability and flame retardancy, which were mainly attributed to the lamellar barrier, gas phase quenching of volatiles containing phosphorus, the catalytic charring of transition metal vanadium, and the synergistic decomposition of oxalic acid structure and phosphorus phase to participate in charring to form a stable char layer.

## 2. Results and Discussion

### 2.1. Structural Analysis of AVOPh

To characterize the structure of AVOPh, XRD, FTIR, TGA, and FESEM tests were carried out. As shown in Figure 1a, XRD results showed that AVOPh showed good crystallinity, and characteristic diffraction peaks appeared at 2θ = 13.2°, 26.4°, and 40.5°, corresponding to (100), (200), and (300) diffraction planes, respectively, indicating that AVOPh had a layered structure [23,24]. The characteristic peaks of VPh at 2θ = 11.9° and 24.0° correspond to (001) and (002) diffraction planes, respectively [24]. In the FTIR spectrum of Figure 1b, the absorption peaks near 3521 cm^−1^ and 3166 cm^−1^ are attributed to the vibration of N-H. The tensile vibration of V = O and V-O appears at 647 cm^−1^ and 523 cm^−1^. The characteristic absorption bands of P-O appear at 1152 cm^−1^ and 952 cm^−1^ [25,26]. In addition, C-O bands at 1651 cm^−1^ and 900 cm^−1^ are assigned to the oxalate structure, and the peak at 1445 cm^−1^ corresponds to the O-H bond [26]. The TGA was used to investigate the thermal stability of AVOPh, AVPh, VPh, and pure EP. From Figure 1c, the first stage of mass loss of AVOPh decreased by 8.2 wt% before 250 °C, mainly due to the desorption of absorbed water and the decomposition of a small amount of amine. The second significant weight loss occurs at 280 °C, mainly due to the thermal decomposition of the AVOPh structure. The mass loss is 21.5 wt% in the range of 280~500 °C. The initial decomposition temperature (T_5%_) of pure EP is 360 °C. From the results, the main decomposition temperature of AVOPh and EP is similar. After thermal decomposition, the final residual of AVOPh is 69.4 wt% at 700 °C. AVPh with a final residual of 62.5 wt% was less stable compared with AVOPh. The mass loss of VPh decreases by 17 wt% occurred before 150 °C, presumably due to the loss of adsorbed water, and the final residual was 82.5 wt% at 700 °C. The main decomposition temperatures of AVOPh and AVPh are lower than that of EP. According to the results of thermal analysis, the decomposition temperature of AVOPh is slightly lower than that of EP. AVOPh is suitable as a flame retardant for EP. Figure 1d–f are the SEM images of AVOPh, AVPh, and VPh, respectively. It can be seen that AVOPh, AVPh, and VPh are all layered structures, which is beneficial for the EP/AVOPh composites to play a lamellar barrier role in the flame retardant process.

### 2.2. Thermal Stability of EP Composites

To analyze the thermal stability of EP composites, a TGA measurement was carried out. The TGA curves of samples in an N_2_ atmosphere are shown in Figure 2. The specific values are listed in Table 1. From the results, the initial decomposition temperature (T_5%_) of EP is 360 °C, and the initial decomposition temperature of EP/AVOPh composites is decreased. The T_5%_ of EP/8 wt% AVOPh composites are reduced to 339 °C, indicating that the vanadium group has a catalytic effect on EP. In addition, the temperature of 50% weight loss (T_50%_) and maximum decomposition temperature (T_max_) of EP/AVOPh composites also decrease. After 280 °C, the decomposition of AVOPh makes the transition metal vanadium, oxalic acid structure, and phosphorus participate in the reactions of the condensation phase in the combustion, promoting the further char formation of the EP matrix. At 700 °C, the residue of EP is 15.3%. When the contents of AVOPh are 2, 4, 6, and 8 wt%, the residual yield of EP/AVOPh composites increases to 19.8%, 21.2%, 21.7%, and 23.0%, respectively. The results indicate that the addition of AVOPh improves the high-temperature thermal stability and residual yield of EP/AVOPh composites, mainly due to the catalytic capacity of the transition metal vanadium and the phosphorus compounds, which promote EP decomposition and cross-linking of decomposition products into a char layer [27]. Notably, the residue (22.6%) of EP/4 wt% VPh composites at 700 °C is higher than that of EP/4 wt% AVOPh composites, which is mainly due to the higher content of vanadium and phosphorus in VPh, exhibiting better catalytic ability [28]. The residual content of EP/4 wt% AVPh composites is only 17.8%, indicating that the existence of an oxalic acid structure can be beneficial to improve the residual char content and high thermal stability of EP composites.

### 2.3. Flame Retardancy of EP Composites

Promoting the self-extinguishing of combustible materials after ignition has potential value for reducing the fire hazard of polymers [29]. To explore the influence of AVOPh on the flame retardant performance of EP composites, vertical combustion and limiting oxygen index (LOI) tests were carried out. The results are listed in Table 2 and Figure 3. The UL-94 vertical combustion test shows that EP is no rating (NR). When 4, 6, and 8 wt% AVOPh are added, EP/AVOPh composites can reach the UL-94 V1 level. Specifically, the (t_1_ + t_2_) value of EP/6 wt% AVOPh composites is only 16 s, which is close to the UL-94 V0 level. As the content of AVOPh continues to increase, the (t_1_ + t_2_) value of EP/8 wt% AVOPh composites increases again, mainly due to the uneven dispersion of AVOPh in the EP matrix [30]. From the LOI results in Figure 4 and Table 2, the flame retardancy of EP composites was improved when the AVOPh was added to EP. When 2 wt% AVOPh is added to the EP matrix, the LOI value increases from 25.9% to 30.1%. After the addition of 4, 6, and 8 wt% AVOPh, the LOI values of EP/AVOPh composites increase to 31.7%, 32.8%, and 34.2%, respectively. The above results have positive significance for improving the fire safety of EP/AVOPh composites. This is the effect of AVOPh combined with gas phase and condensed phase flame retardancy during EP composite combustion. In comparison, although the EP/4 wt% AVPh composites can reach the V1 level, the (t_1_ + t_2_) value is as high as 41 s; EP/4 wt% VPh composites still do not have a rating of UL-94. Additionally, the LOI values of EP/4 wt% AVPh and EP/4 wt% VPh composites are 30.2% and 29.8%, respectively, which are inferior to that of EP/4 wt% AVOPh composite. It can be concluded that the presence of amino groups plays a positive role in promoting flame extinction. The improved flame retardancy of EP/AVOPh composites is mainly attributed to the release of phosphorus free radicals, NH_3_, and water vapor in the gas phase during the decomposition process of AVOPh. They dilute the concentration of combustible and oxygen around the combustion matrix and promote flame extinction. In addition, the dense and strong char layer can isolate the external heat source and accelerate EP/AVOPh composite self-quenching [31,32].

To further analyze the flame retardancy of EP/AVOPh composites, a cone calorimeter test (CCT) was carried out. Heat release rate (HRR), total smoke production (TSP), CO production (COP), and CO_2_ production (CO_2_P) are shown in Figure 4, and the specific values are listed in Table 3. The peak heat release rate (PHRR) of pure EP is 1004 kW·m^−2^, indicating EP has great thermal hazard. The addition of AVOPh significantly reduces the PHRR values of EP composites. When 2, 4, 6, and 8 wt% AVOPh are added, the PHRR values of EP/AVOPh composites decrease to 931, 796, 795, and 675 kW·m^−2^, respectively. The PHRR value of EP/8 wt% AVOPh composites decreases by 32.7% compared with that of pure EP. The PHRR values of EP/4 wt% AVPh and EP/4 wt% VPh composites are 877 kW·m^−2^ and 840 kW·m^−2^, which are higher than that of EP/4 wt% AVOPh composites. Smoke production and toxicity are important indicators of fire hazard. The TSP value, peak CO production (PCOP), and peak CO_2_ production (PCO_2_P) of EP are 21.2 m^2^, 0.035 g·s^−1^, and 0.54 g·s^−1^, respectively, indicating EP has a large production of smoke and toxic gas. When EP/AVOPh composites with 2, 4, 6, and 8 wt% AVOPh, TSP values decrease to 17.6, 17.4, 16.8, and 16.4 m^2^, PCOP values decrease to 0.031, 0.026, 0.024, and 0.022 g·s^−1^, and PCO_2_P values decrease to 0.52, 0.44, 0.42, and 0.36 g·s^−1^, respectively. Compared with EP, the TSP, PCOP, and PCO_2_P values of EP/8 wt% AVOPh are decreased by 20.4%, 37.1%, and 33.3%, respectively. The results indicate that AVOPh can suppress smoke production and toxic gas release in EP composites, which provides more favorable conditions for rescue and escape in the fire. This can be attributed to the lamellar barrier and excellent charring of layered AVOPh, which act as heat insulation and gas isolation. Secondly, the phosphorus free radicals in the gas phase also play a key role in reducing the emission of smoke and toxic gases. In addition, the synergistic participation of oxalic acid structure and ammonium in the carbonization process forms a high-quality expanded char layer, thereby providing thermal insulation and smoke suppression [33,34].

To further confirm the positive influence of AVOPh on the char-forming performance of EP, the mass loss during the CCT is shown in Figure 5. The specific values are listed in Table 3. The mass residual of pure EP after the CCT is 18.7%. When the contents of AVOPh are 2, 4, 6, and 8 wt%, the residual amounts of EP/AVOPh composites are increased to 25.2%, 30.3%, 29.9%, and 37.3%. In addition, the residual amounts of EP/4 wt% AVPh composites and EP/4 wt% VPh composites are only 25.0% and 24.3%, indicating that the oxalic acid structure is involved in char formation during the decomposition of EP. The possible reason is that AVOPh promotes the formation of more expansive char on the surface of EP at the beginning of combustion, thereby isolating oxygen and heat from entering the EP interior. It provides a sufficient time of contact for small molecules decomposed from EP and vanadium, phosphides, and oxalic acid. In a word, AVOPh promotes the char formation of EP composites and improves the quality of the char layer after combustion, which further effectively isolates the internal and external mass exchange, improving the flame retardant performance of EP/AVOPh composites [35,36].

To observe the charring effect of AVOPh in EP composites, the SEM images of the inner and outer residue of pure EP and EP/AVOPh composites are shown in Figure 6. The outer and inner char of pure EP is shown in Figure 6a,b. The outer char has obvious collapse, with large and dense cracks. During the combustion process of the EP, it cannot act as a barrier during the combustion. There are large pores and cracks in the internal char. This is mainly caused by the intense combustion of EP and the rapid escape of a large amount of heat and smoke within a short time. The residue of EP/2 wt% AVOPh composites is shown in Figure 6c,d. The quality of the external char is obviously improved. Cracks of external char disappear, and the char is dense. This indicates that AVOPh plays a certain barrier role. At high temperatures, the transition metal vanadium and phosphorus compounds catalyze the dehydration and charring of the EP matrix, accelerating the formation of the char layer [37]. The large pores and cracks in the internal char are reduced, indicating that the intensity of combustion is reduced compared to pure EP. The residue of the EP/6 wt% AVOPh composites is shown in Figure 6e,f. The outer char is very dense and compact with a smooth surface. Although the outer char of the EP/6 wt% AVOPh composites becomes dense, the surface is relatively loose, which is crucial for improving the barrier property of the char. The inner char becomes thicker compared to that of EP/2 wt% AVOPh composites. The results indicate that AVOPh plays a significant catalytic role in forming compact char, improving the flame retardancy of EP/AVOPh composites [38].

### 2.4. Flame Retardant Mechanism of AVOPh in EP

According to the above analysis, the flame retardant mechanism of EP/AVOPh composites is discussed in Figure 7. AVOPh can promote flame extinguishment and inhibit heat release and smoke production, which improves the flame retardancy of EP/AVOPh composites. This can be attributed to the two-dimensional layered structure of AVOPh acting as a physical barrier effect during combustion and limiting energy transfer and gas escape [39,40,41]. Secondly, in the process of gas-phase flame retarding, AVOPh releases crystal water and NH_3_, which dilute the concentration of combustible volatiles and oxygen around the combustion matrix. After the decomposition of the phosphorus compound, AVOPh releases HPO· and PO· free radicals to further capture and dilute high-energy free radicals, interfering with the combustion reaction and promoting flame self-extinguishing [42,43,44]. In addition, the transition metal vanadium oxidizes to form V_2_O_5_ and catalyzes the matrix to form char in the condensed phase. Meanwhile, oxalic acid structure decomposition participates in the char formation in this process. The quality of the char layer can be enhanced by the condensation of vanadium-based phosphate compounds to form H_3_PO_4_, (VO)_2_P_2_O_7_, and H_4_P_2_O_7_, leading to forming a dense and strong char layer to protect the internal unburned matrix and prevent further mass exchange [45,46].

## 3. Experimental Section

### 3.1. Materials

Phosphoric acid (H_3_PO_4_, 85% solution in H_2_O), vanadium pentoxide (V_2_O_5_), ammonia solution (NH_4_OH 30 wt% aqueous solution), and oxalic acid dihydrate (C_2_H_2_O_4_·2H_2_O) were purchased from Sinopharm Chemical Reagent Co., Ltd. (Shanghai, China). Epoxy resin (NPEL 128) was purchased from NanYa Electronic Materials Co., Ltd. (Kunshan, China). 4, 4′-diaminodiphenyl methane (DDM) was purchased from Jiacheng Materials Co., Ltd (Dongguan, China).

### 3.2. Synthesis of AVOPh, AVPh, and VPh

Synthesis of AVOPh((NH_4_)_2_[VO(HPO_4_)]_2_(C_2_O_4_)·5H_2_O): First, 6 mL deionized water and 0.6 mL H_3_PO_4_ were added to a 50 mL beaker. Then, 0.18 g V_2_O_5_ and 0.26 g C_2_H_2_O_4_·2H_2_O were added to the beaker. Then, 2.3 mL NH_4_OH was dropped into the beaker and stirred continuously for 10 min. Finally, the above substances were transferred into a 50 mL Teflon reactor at 140 °C for three days. After the reaction, the green products were collected by vacuum extraction and filtrated, washed with deionized water three times, then dried in an oven at 80 °C. The product was labeled as AVOPh.

Synthesis of AVPh((NH_4_)VOPO_4_·1.5H_2_O): First, 0.90 g V_2_O_5_, 1.29 g C_2_H_2_O_4_·2H_2_O, 2.1 mL H_3_PO_4_, 2.6 mL NH_4_OH and 3 mL deionized water were added to a 100 mL beaker and stirred continuously for 10 min to make the various substances fully and evenly mixed. Then, the mixture was poured into a Teflon reactor with a capacity of 50 mL and placed in the oven at 140 °C for three days. After the reaction was completed, the green product was obtained. The product was dried and washed in the same way and labeled as AVPh.

Synthesis of VPh(VOPO_4_·2H_2_O): First, 115.4 mL deionized water was added into the 250 mL round-bottomed flask. Then 26.6 mL H_3_PO_4_ was slowly added into the round-bottomed flask and stirred at the speed of 500 rpm. Finally, 4.8 g V_2_O_5_ was added to the above solution. The round-bottomed flask was placed in an oil bath and refluxed at 110 °C for 16 h. The product was dried and washed in the same way and labeled as VPh.

### 3.3. Preparation of EP/AVOPh Composites

Firstly, AVOPh was uniformly dispersed in acetone through ultrasonicated dispersion. A total of 20 mL acetone is required per gram of AVOPh. Then, epoxy resin was added to the above dispersion liquid and the EP/AVOPh composites were prepared via ultrasonicated dispersion. The addition of AVOPh was controlled in 2, 4, 6, and 8 wt% in EP composites. Then the beaker was placed in the oil bath at 90 °C and stirred at 300 rpm for 4 h to remove the acetone. Curing agent DDM (EP:DDM = 4:1) was added and stirred continuously until DDM at the speed of 300 rpm. The mixture was put in a vacuum oven to remove bubbles at 80 °C for 5 min. Then the above-mentioned mixture was poured into preheated molds as quickly as possible. The samples were cured at 110 °C/2 h, 130 °C/2 h, and 150 °C/2 h, respectively. To further clarify the flame retardant effect of each component of AVOPh in epoxy resin, EP/4 wt% AVPh composites and EP/4 wt% VPh composites were prepared via the same method as comparative samples of EP/AVOPh. The formula of the EP/AVOPh composite is listed in Table 4.

### 3.4. Characterization

X-ray diffraction (XRD) analysis was performed using Rigaku’s MAX-RB diffractometer with a range of 5° to 80° at an operating voltage of 40 kV. Infrared spectroscopic characterization was conducted using 6700 spectrometers from Nicolet, USA, with the KBr tablet method in the range of 400 to 4000 cm^−1^. The field emission scanning electron microscope (SEM) was characterized with an EVO MA15 scanning electron microscope from Zeiss Company, Aalen, Germany. The thermogravimetric analysis of materials was performed using a TG50 thermal analyzer (the same sample was tested two times, and the repeatability was good). The linear heating rate was 10 °C·min^−1^ under N_2_. The limit oxygen index test (LOI) was performed using a JF-3 oxygen index meter from Jiangning Analytical Instrument Co., Ltd. (Nanjing, China) with a spline size of 130 × 6.5 × 3.2 mm^3^ according to ASTM D 2863. The vertical combustion test (UL-94) was also carried out using a vertical combustion apparatus from Jiangning Analytical Instrument Co., Ltd. The spline size was 130 × 12.7 × 3.2 mm^3^, and the test standard was carried out according to ASTM D 3801. The same sample was tested 15 times by LOI and vertical combustion. The cone calorimeter is an instrument from Stanton Redcroft, London, UK, with a continuous heat flux of 50 kW·m^−2^, and the sample was wrapped in aluminum foil with a size of 100 × 100 × 3 mm^3^. The same sample was tested three times with CCT.

## 4. Conclusions

In this work, layered AVOPh was synthesized using the hydrothermal method, which was used to prepare EP/AVOPh composites. The structure–activity relationship of phosphorus and nitrogen and the role of an oxalic acid structure in the flame retardancy of EP/AVOPh composites were investigated with TGA, LOI, vertical combustion, and CCT. Compared with pure EP, the residue of EP/8 wt% AVOPh composites increased from 15.3% to 23.0% at 700 °C in an N_2_ atmosphere. The combustion results showed that the EP/4 wt% AVOPh composites passed a V1 rating, and the LOI value of EP/8 wt% AVOPh composites increased from 25.9% to 34.2% compared with pure EP. The PHRR, TSP, PCOP, and PCO_2_P values of EP/8 wt% AVOPh composites decreased by 32.7%, 26.8%, 37.1%, and 33.3%, respectively. After the CCT, the residue of EP/4 wt% AVOPh composites was high at 30.3%, compared with 25.0% and 24.3% of EP/4 wt% AVPh and EP/4 wt% VPh composites, indicating that the oxalic acid structure was involved in the char formation during the combustion. The improved flame retardancy was mainly due to the synergistic effect of different components in AVOPh. In the condensed phase, the two-dimensional layered structure of AVOPh acted as a physical barrier effect during combustion and limited energy transfer and gas escape. Phosphoric acid and oxalic acid promoted the dehydration and carbonization of EP. The transition metal vanadium oxidized the catalyzed decomposition product of EP to form char. In the gas phase, water and NH_3_ diluted combustible gas are released by AVOPh. The decomposition of AVOPh released HPO· and PO· free radicals to further capture and dilute high-energy free radicals, interfering with the combustion reaction and promoting flame self-extinguishing. 

## Figures and Tables

**Figure 1 molecules-28-03322-f001:**
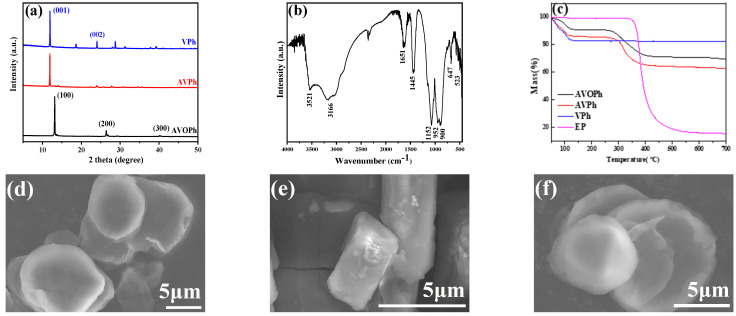
(**a**) XRD pattern of AVOPh, AVPh, and VPh; (**b**) FTIR spectrum of AVOPh; (**c**) TGA curves of AVOPh, AVPh, VPh, and EP; (**d**–**f**) SEM images of AVOPh, AVPh, and VPh.

**Figure 2 molecules-28-03322-f002:**
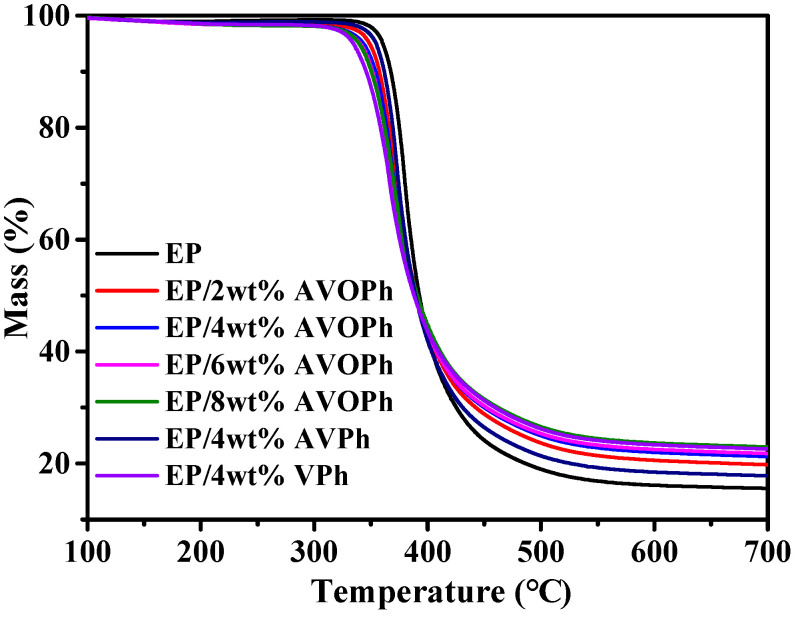
TGA curves of pure EP and its composites.

**Figure 3 molecules-28-03322-f003:**
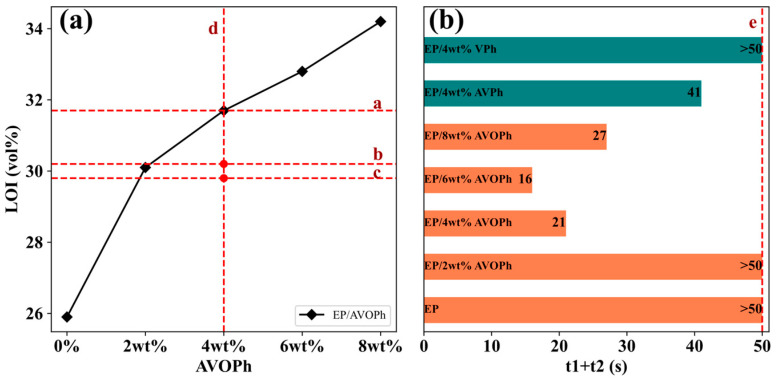
The LOI and vertical combustion curves of EP and EP composites: (**a**) LOI; (**b**) vertical combustion. a: the LOI of EP/4 wt% AVOPh. b: the LOI of EP/4 wt% AVPh. c: the LOI of EP/4 wt% VPh. d: additive ratio. e: V1: UL-94 V1 (10 s < t_1_ + t_2_ < 50 s, no dripping or dripping does not ignite cotton).

**Figure 4 molecules-28-03322-f004:**
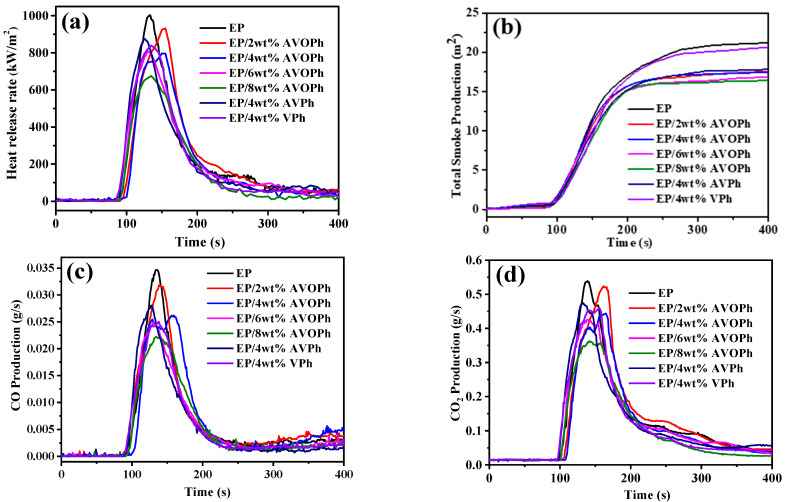
CCT curves of pure EP and its composites: (**a**) HRR curves; (**b**) TSP curves; (**c**) COP curves; (**d**) CO_2_P curves.

**Figure 5 molecules-28-03322-f005:**
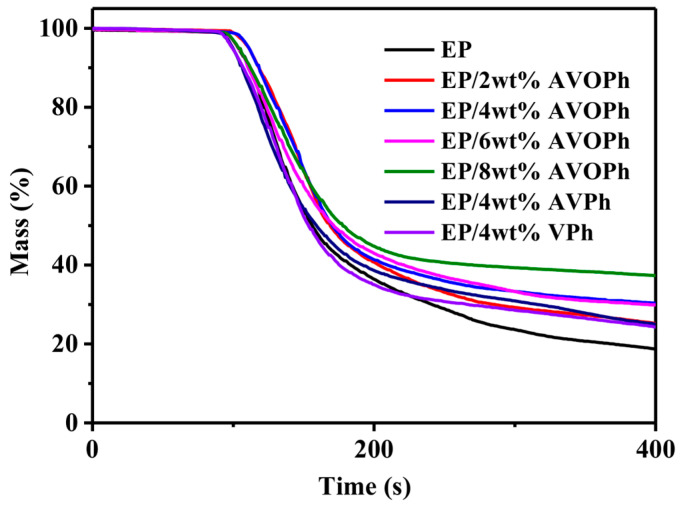
Mass loss curves of pure EP and its composites at CCT.

**Figure 6 molecules-28-03322-f006:**
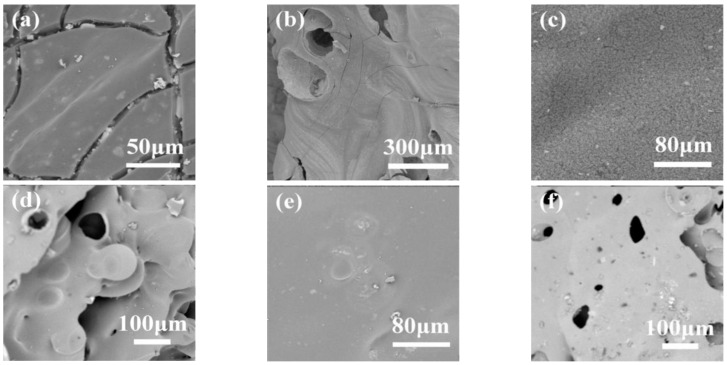
SEM images of char layer: (**a**) external and (**b**) internal char layer of pure EP; (**c**) external and (**d**) internal char layer of EP/2 wt% AVOPh composites; (**e**) external and (**f**) internal char layer of EP/6 wt% AVOPh composites.

**Figure 7 molecules-28-03322-f007:**
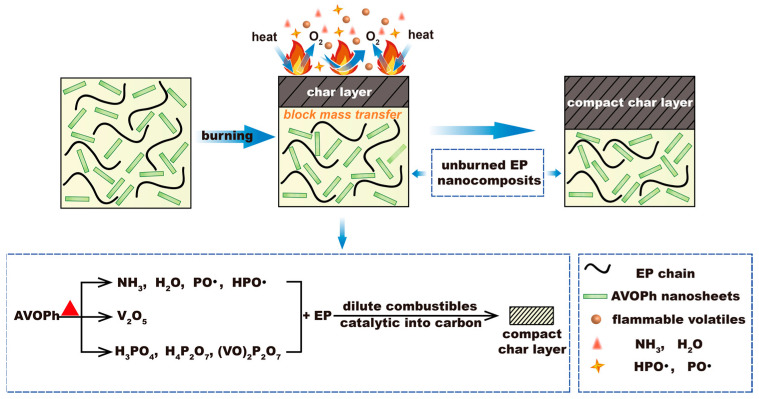
Schematic illustration of the mechanism for the enhanced flame resistance of EP/AVOPh composites.

**Table 1 molecules-28-03322-t001:** TGA data of pure EP and its composites.

Samples	T_5%_ (°C)	T_50%_ (°C)	T_max_ (°C)	Residues (wt%, 700 °C)
EP	360	394	380	15.3
EP/2 wt% AVOPh	349	391	374	19.8
EP/4 wt% AVOPh	343	390	369	21.2
EP/6 wt% AVOPh	340	389	368	21.8
EP/8 wt% AVOPh	339	391	369	23.0
EP/4 wt% AVPh	355	389	370	17.8
EP/4 wt% VPh	334	388	366	22.6

**Table 2 molecules-28-03322-t002:** UL-94 and LOI results of EP and its composites.

Samples	LOI (vol%)	UL-94	
		t_1_ + t_2_ (s)	Rating
EP	25.9	>50	NR *
EP/2 wt% AVOPh	30.1	>50	NR
EP/4 wt% AVOPh	31.7	21	V1 **
EP/6 wt% AVOPh	32.8	16	V1
EP/8 wt% AVOPh	34.2	27	V1
EP/4 wt% AVPh	30.2	41	V1
EP/4 wt% VPh	29.8	>50	NR

* NR: no rating of UL-94 (t_1_ + t_2_ > 50 s). ** V1: UL-94 V1 (10 s < t_1_ + t_2_ < 50 s, no dripping or dripping does not ignite cotton).

**Table 3 molecules-28-03322-t003:** CCT data of pure EP and EP composites.

Samples	PHRR (kW/m^2^)	TSP (m^2^)	PCOP (g/s)	PCO_2_P (g/s)	Mass Residue (%)
EP	1004	22.4	0.035	0.54	18.7
EP/2 wt% AVOPh	931	17.6	0.031	0.52	25.2
EP/4 wt% AVOPh	796	17.4	0.026	0.44	30.3
EP/6 wt% AVOPh	795	16.8	0.024	0.42	29.9
EP/8 wt% AVOPh	675	16.4	0.022	0.36	37.3
EP/4 wt% AVPh	877	17.8	0.028	0.47	25.0
EP/4 wt% VPh	840	19.7	0.025	0.45	24.3

**Table 4 molecules-28-03322-t004:** Ingredients of EP composites.

Samples	Components			
	EP (wt%)	AVOPh (wt%)	AVPh (wt%)	VPh (wt%)
EP	100	0	0	0
EP/2 wt% AVOPh	98	2	0	0
EP/4 wt% AVOPh	96	4	0	0
EP/6 wt% AVOPh	94	6	0	0
EP/8 wt% AVOPh	92	8	0	0
EP/4 wt% AVPh	96	0	4	0
EP/4 wt% VPh	96	0	0	4

## Data Availability

Not applicable.

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
