# Peer review of "Effect of Layered Aminovanadic Oxalate Phosphate on Flame Retardancy of Epoxy Resin"

_molecules, 2023, doi:10.3390/molecules28083322_

Round 1

Reviewer 1 Report

Comments to the Authors
The authors prepared layered ammonium vanadium oxalate-phosphate (AVOPh) as flame retardants for epoxy resins. EP/AVOPh composites exhibit significantly better UL-94 tests compared to EP, the LOI values were increased and according to the CCT test
the peak heat release rate (PHHR), total smoke production (TSP), the peak of CO production (PCOP) and peak of CO2 production (PCO2P) decreased by 32.7%, 26.7%, 37.1% and 33.3% for EP/AVOPh composites comparing with that of EP. The structure-activity relationship of phosphorus, nitrogen and the role of oxalic acid structure in flame retardancy were investigated.

1.      A scientific paper contains many grammatical errors. I ask the authors to check and correct the English in detail, especially in the first part of the article (Experimental section).

2.      Page 4, line 139 – the mass loss at 500 °C for AVOPh is almost 30 wt% according to the TG curve.

3.      Please add also TG curve of EP (Figure 1c).

4.      What are the dimensions of AVOPh according to SEM images? Do you think that the size of AVOPh can affect the properties, given that AVPh and VPh are larger dimensions?

5.      Page 6, Table 3 – please explain NR and V1 to readers.

Author Response

For Reviewer #1

  1. A scientific paper contains many grammatical errors. I ask the authors to check and correct the English in detail, especially in the first part of the article (Experimental section).

Replying: We check and correct the English in detail. Many grammar and expressions have been revised.

  1. Page 4, line 139 – the mass loss at 500 °C for AVOPh is almost 30 wt% according to the TG curve.

Replying: Change the expression to ‘The mass loss was 21.5wt% from 280 oC to 500 oC’.

  1. Please add also TG curve of EP (Figure 1c).

Replying: TG curve of EP was added.

  1. What are the dimensions of AVOPh according to SEM images? Do you think that the size of AVOPh can affect the properties, given that AVPh and VPh are larger dimensions?

Replying: To better understand the size of AVOPh, an image of AVOPh has been replaced in the article. The dimensions of AVOPh is about 5-10um in SEM images. The sizes of AVOPh, AVPh and VPh are at the same level, with no significant difference. The effect of size on composites is not considered in this paper.

  1. Page 6, Table 3 – please explain NR and V1 to readers.

Replying: The explanations for V and NR are placed below Table 3.

*NR: no rating of UL-94(t1+t2>50s).

**V1: UL-94 V1( t1+t2<50s, no dripping or dripping does not ignite absorbent cotton).

Reviewer 2 Report

Please find my comments on the article entitled Effect of Layered Aminovanadic Oxalate Phosphate on Flame Retardancy of Epoxy Resin by Po hu et al.

The manuscript is based on an old idea I didn’t find anything novel and interesting in this article. My further comments are as follows

Abstract:

The aim of the study is not clear. Abstract needs substantial improvement.

Introduction:

Introduction needs extensive revisions and improvement. The authors should revise the introduction part by adding latest literature and new paragraphs.

The introduction is not enough its very short.

Materials and Methods:

Materials and methods are not well defined.

Its based on the same old methodologies. New methodologies or new protocol should be added to the revised manuscript.

Discussion :

Discussion section is very well written but still it needs improvement. English language of the whole manuscript should be revised.

Figures quality is good.

At this stage I would recommend major revisions.

Author Response

Abstract: The aim of the study is not clear. Abstract needs substantial improvement.

Replying: The aim of the study has been added to the abstract section: To alleviate the fire hazard of epoxy resin (EP), layered ammonium vanadium oxalate-phosphates (AVOPh) with the structural formula of (NH4)2[VO(HPO4)]2(C2O4).5H2O was prepared by one-step hydrothermal method and mixed into EP matrix to prepare EP/AVOPh composites.

  1. Introduction: Introduction needs extensive revisions and improvement. The authors should revise the introduction part by adding latest literature and new paragraphs. The introduction is not enough very short.

Replying: The introduction part by adding latest literature and new paragraphs was revisions and improvement.

  1. Materials and Methods: Materials and methods are not well defined. It is based on the same old methodologies. New methodologies or new protocol should be added to the revised manuscript.

  Replying: Materials are well defined in the study.

 ammonium vanadium oxalate-phosphates: (AVOPh(NH4)2[VO(HPO4)]2(C2O4)·5H2O),

 ammonium vanadium phosphates: AVPh((NH4)VOPO4·1.5H2O)

vanadium phosphates: VPh(VOPO4·2H2O)

The synthesis methods of materials are relatively mature. In this paper, hydrothermal synthesis is mainly used for the synthesis of flame retardant materials. The synthesis of composites are mainly ultrasonicated dispersion.

  1. Discussion: Discussion section is very well written but still it needs improvement. English language of the whole manuscript should be revised.

Replying: Discussion section is improved. English language of the whole manuscript has been revised.

Reviewer 3 Report

The manuscript entitled, “Effect of Layered Aminovanadic Oxalate Phosphate on Flame Retardancy of Epoxy Resin will benefit the researchers working on Epoxy resins. The manuscript needs a few amendments before considering for publication in Molecules. Therefore, major revision is recommended concerning the quality of English/Grammar and scientific content. The suggested changes are as follows:

Abstract

Add a concluding statement to give a future direction to the outcome or for the researchers working in this area.

Keywords

Provide two more keywords

Introduction

Can be more informative

Add the market size of Epoxy resins

Experimental section

Line 86: How much acetone and AVOPh were added?

Line 86: Not ‘Ultrasonic’, it should be ‘ultrasonicated’, also provide the frequency at which it was ultrasonicated.

Line 89: Not ‘Ultrasonic’, it should be ‘ultrasonicated’, also provide the frequency at which it was ultrasonicated.

Line 90: Stirred at what rpm?

Line 91: Stirred at what rpm?

Line 94: ‘Then pour the above system’ is grammatically incorrect. It should be ‘the above-mentioned mixture was poured’.

Please work on the English language throughout the manuscript.

I don’t see any statistical analysis statement in the experimental section

What was the number of replicates?

Authors should rewrite the experimental section for it contains a lot of scientific and grammatical errors.

Result and discussion

Result is lacking statistical relevance, for instance, was the TG so accurate? Didn’t the TG fluctuate between the replicates?

Repeated use of ‘In order to’, please rephrase it just to ‘To’

Conclusion

There should be space between numerical data and their units

Provide a take-home statement at the end of the conclusion

The manuscript needs work with English/Grammar and statistical analysis.

The data availability statement is missing.

Author Response

For Reviewer #3

  1. Abstract: Add a concluding statement to give a future direction to the outcome or for the researchers working in this area.

Replying: A concluding statement to give a future direction in the abstract: AVOPh is expected to serve as a new high-efficiency flame retardant for EP.

  1. Keywords: Provide two more keywords

Replying: Provide two more keywords: Hydrothermal method; Layered structure; Phosphate; Flame retardancy; vanadium; Epoxy resin

  1. Introduction: Can be more informative, Add the market size of Epoxy resins

Replying: The introduction part by adding more informative and the market size of epoxy resins.

In the past decade, the global production of epoxy resin increased from 2.96 million tons in 2017 to 3.73 million tons in 2021, with a compound annual growth rate of 5.95%. As of 2021, the global total production capacity of epoxy resin is 5.37 million tons.

An introduction to the flame retardant synergistic effects of phosphorus nitrogen and phosphorus and metal compounds is added to the introduction section, as following:

The synergistic effect of nitrogen and phosphorus has a phosphorus nitrogen structure activity relationship, diluting the concentration of combustibles and oxygen around the combustion substrate, and promoting flame extinction [20, 21]. The results indicated that adding 7wt% nitrogen/phosphorus modified lignin to epoxy resin achieved UL-94 V0 grade and limiting oxygen index of 31.4%, respectively. In addition, the residual char became more dense [22]. NiNH4PO4·H2O nanoflakes have good flame retardancy in EP. When the addition amount was 5%, the peak heat release rate and peak smoke production of the composites were 69.1% and 36.5% lower than those of pure EP, respectively. NiNH4PO4·H2O promoted the formation of a stable carbon layer and released noncombustible gases. It prevented heat and oxygen transfer, and diluted the concentration of combustible gases [21].

  1. Experimental section

Line 86: How much acetone and AVOPh were added?

Replying: AVOPh were is uniformly dispersed in acetone through ultrasonic dispersion. 20 ml acetone is required per gram of AVOPh.  

  1. Line 86: Not ‘Ultrasonic’, it should be ‘ultrasonicated’, also provide the frequency at which it was ultrasonicated. Line 89: Not ‘Ultrasonic’, it should be ‘ultrasonicated’, also provide the frequency at which it was ultrasonicated.

Replying: Ultrasonic’ was replaced by ultrasonicated in the part of 2.3 Preparation of EP/AVOPh composites.

  1. Line 90: Stirred at what rpm? Line 91: Stirred at what rpm?

Replying: After that, curing agent DDM (EP:DDM=4:1) was added and stirred continuously until DDM was completely dissolved at the speed of 300rpm.

  1. Line 94: ‘Then pour the above system’ is grammatically incorrect. It should be ‘the above-mentioned mixture was poured’.

Replying: “Then pour the above system” was replaced by ‘the above-mentioned mixture was poured’.

 The mixture was put in a vacuum oven to remove bubbles at 80 oC for 5min. Then the above-mentioned mixture was poured into preheated molds as quickly as possible.

  1. Please work on the English language throughout the manuscript.

Replying: Thanks for your suggestion! After many revisions to the paper, many grammatical and expressive errors have been corrected, specifically in the text.

  1. I don’t see any statistical analysis statement in the experimental section. What was the number of replicates?

Replying: The thermogravimetric analyzer of materials was used by TG50 thermal analyzer (The same sample was tested two times and the repeatability was good.). The same sample was tested for 15 times by LOI and vertical combustion. Test the same sample three times by CONE.

  1. Authors should rewrite the experimental section for it contains a lot of scientific and grammatical errors.

Replying: Thanks for your suggestion! After many revisions to the paper, many grammatical and expressive errors have been corrected. The experimental section was rewrite.

  1. Result and discussion: Result is lacking statistical relevance, for instance, was the TG so accurate? Didn’t the TG fluctuate between the replicates?

Replying: The tests of TG, LOI, vertical combustion and CCT were verified through repeated experiments. The results of TG and CCT were good repeatability. The results of LOI and vertical combustion were average of data.

  1. Repeated use of ‘In order to’, please rephrase it just to ‘To’

Replying: ‘In order to’ was replaced by ‘To’.

  1. Conclusion: There should be space between numerical data and their units. Provide a take-home statement at the end of the conclusion.

Replying: space between numerical data and their units was adjusted. A take-home statement at the end of the conclusion was provide:

The improved flame retardancy was mainly due to the synergistic effect of different components in AVOPh. In the condensed phase, the two-dimensional layered structure of AVOPh played a physical barrier effect during combustion and limited energy transfer and gas escape. Phosphoric acid and oxalic acid promoted the dehydration and carbonization of EP. The transition metal vanadium oxidizes catalyzed decomposition product of EP to form char. In the gas phase, water and NH3 diluted combustible gas released by AVOPh. The decomposition of AVOPh released HPO• and PO• free radicals to further capture and dilute high-energy free radicals, interfering with the combustion reaction, and promoting flame self-extinguishing.

  1. The manuscript needs work with English/Grammar and statistical analysis. The data availability statement is missing.

Replying: Thanks for your suggestion! After many revisions to the paper, many grammatical and expressive errors have been corrected, specifically in the text.

Reviewer 4 Report

In this work, Layered ammonium vanadium oxalate-phosphates (AVOPh) were prepared by one-step hydro-thermal method and mixed into EP matrix to prepare EP/AVOPh flame retardant. But there are still some issues to be resolved. Minor revision are recommended.

1.       Please rearrange the layout of Figure 1. The image is not the same size and is not aligned.

2.       In the analysis of Figure 1c, the author says that "AVOPh is more suitable as a flame retardant for EP", please give the reason more directly.

3.       In Figure 2, the residues of EP/AVOPh composites increase almost linearly as the amount of AVOPh increases. Is this simply because AVOPh does not decompose?

4.       The authors refer to "a high quality carbon layer" in all their analyses and suggest reducing the frequency of this reason. Or by observing the carbon layer through SEM or other methods to convince the reader.

5.       The data of many tables in the article can be presented in bar charts, which will make it easier for readers to accept the information.

6.       Please refine the grammar.

Author Response

  1. Please rearrange the layout of Figure 1. The image is not the same size and is not aligned.

Replying: Figure 1 has been adjusted.

Figure 1. (a) XRD pattern of AVOPh, AVPh and VPh; (b) FTIR spectrum of AVOPh; (c) TGA curves of AVOPh, AVPh, VPh and EP; (d-f) SEM images of AVOPh, AVPh and VPh.

  1. In the analysis of Figure 1c, the author says that "AVOPh is more suitable as a flame retardant for EP", please give the reason more directly.

Replying: According to the results of thermal analysis, the decomposition temperature of AVOPh and AVPh is slightly lower than that of EP. AVOPh and AVPh is suitable as flame retardant for EP.

  1. In Figure 2, the residues of EP/AVOPh composites increase almost linearly as the amount of AVOPh increases. Is this simply because AVOPh does not decompose?

Replying: The increase in residue comes from two sources. One is the residue of AVOPh. And the other is the carbonization of EP decomposition products under the catalytic carbonization of phosphorus and vanadium.

  1. The authors refer to "a high quality carbon layer" in all their analyses and suggest reducing the frequency of this reason. Or by observing the carbon layer through SEM or other methods to convince the reader.

Replying: SEM images of char layer were added in Figure 6.

Figure 6. SEM images of char layer: (a) external and (b) internal char layer of pure EP; (c) external and (d) internal char layer of EP/2wt% AVOPh composites; (e) external and (f) internal char layer of EP/6wt% AVOPh composites.

  1. The data of many tables in the article can be presented in bar charts, which will make it easier for readers to accept the information.

Replying: The data in this article have corresponding line diagrams except for the data in Table 2. In the article, we have added Figure 3 as the corresponding line diagram in Table 2

Figure 3. The LOI and vertical combustion curves of EP and EP composites: (a) LOI; (b) vertical combustion

  1. Please refine the grammar.

Replying: Thanks for your suggestion! After many revisions to the paper, many grammatical and expressive errors have been corrected, specifically in the text.

Round 2

Reviewer 2 Report

The authors have revised the manuscript and addressed my comments. The article can be accepted for publication.